# Predicting perinatal health outcomes using smartphone-based digital phenotyping and machine learning in a prospective Swedish cohort (Mom2B): study protocol

Ayesha M Bilal [ORCID] ,[1,2] Emma Fransson,[3,4] Emma Bränn,[3] Allison Eriksson,[2,3] Mengyu Zhong,[2,5] Karin Gidén,[3] Ulf Elofsson,[1,3] Cathrine Axfors,[3] Alkistis Skalkidou,[3] Fotios C Papadopoulos[1]

AS and FCP contributed equally.

For numbered affiliations see end of article.

**Correspondence to**
Ayesha M Bilal;
ayesha.bilal@neuro.uu.se

## ABSTRACT

**Introduction** Perinatal complications, such as perinatal depression and preterm birth, are major causes of morbidity and mortality for the mother and the child. Prediction of high risk can allow for early delivery of existing interventions for prevention. This ongoing study aims to use digital phenotyping data from the Mom2B smartphone application to develop models to predict women at high risk for mental and somatic complications.

**Methods and analysis** All Swedish-speaking women over 18 years, who are either pregnant or within 3 months postpartum are eligible to participate by downloading the Mom2B smartphone app. We aim to recruit at least 5000 participants with completed outcome measures. Throughout the pregnancy and within the first year postpartum, both active and passive data are collected via the app in an effort to establish a participant's digital phenotype. Active data collection consists of surveys related to participant background information, mental and physical health, lifestyle, and social circumstances, as well as voice recordings. Participants' general smartphone activity, geographical movement patterns, social media activity and cognitive patterns can be estimated through passive data collection from smartphone sensors and activity logs. The outcomes will be measured using surveys, such as the Edinburgh Postnatal Depression Scale, and through linkage to national registers, from where information on registered clinical diagnoses and received care, including prescribed medication, can be obtained. Advanced machine learning and deep learning techniques will be applied to these multimodal data in order to develop accurate algorithms for the prediction of perinatal depression and preterm birth. In this way, earlier intervention may be possible.

**Ethics and dissemination** Ethical approval has been obtained from the Swedish Ethical Review Authority (dnr: 2019/01170, with amendments), and the project fully fulfils the General Data Protection Regulation (GDPR) requirements. All participants provide consent to participate and can withdraw their participation at any time. Results from this project will be disseminated in international peer-reviewed journals and presented in relevant conferences.

### Strengths and limitations of this study

► The study collects large-scale, temporally sensitive data regarding the user's behaviours in the real world.
► End users' feedback collected allows for app updates and improvements.
► The passive data collection is expected to have lower attrition rate.
► The active data collection is prone to suffer from a higher attrition rate.
► There are high costs associated with recruiting participants and maintaining frontend and backend for the smartphone app.

## INTRODUCTION

Optimal maternal health is important throughout pregnancy, childbirth and the postpartum period to ensure the full potential for the mother, infant and family to get a good start.[1] Two health conditions that are important to address in order to reach a goal of good maternal health are perinatal depression (PND) and preterm birth (PTB), both affecting about every 10th pregnancy worldwide.[2 3]

### Perinatal depression

PND is an episode of major depression with onset anytime during pregnancy and up to 4 weeks postpartum,[4] although in research settings, a period of up to 1 year postpartum is often considered.[5] Antenatal depression affects between 7% and 13% of pregnant women,[6] and postpartum depression (PPD) is estimated to affect between 10% and 20% of all newly delivered mothers,[2]

while many women experience persistent depression throughout the perinatal period.[7] PND is distinct from 'baby blues', which are commonly experienced symptoms of low mood and anxiety that subside within 2 weeks postpartum. PND is both emotionally and physically debilitating like major depression, with additional risks related to the pregnancy and birth, such as PTB, low birth weight, pre-eclampsia and placental abnormalities.[8–10] Moreover, it is associated with retained maternal weight postpartum,[11] decreased breast feeding,[12–14] poor maternal sleep[15] and poor perinatal quality of life.[16] PND can also compromise the critical mother–infant bond, as it affects the mother's caregiving abilities and adaptation to the maternal role,[16 17] and has a long-term impact on the child's cognitive, emotional and behavioural development.[18 19] Furthermore, PND can be characterised by the occurrence of self-harm thoughts, which are linked to long-term somatic and psychiatric morbidity,[20] and increased maternal mortality from suicide in the first year postpartum.[21]

The aetiology of PND is multifactorial, including biological, genetic, psychological and social factors, such as stressful life events, social support, domestic violence, childhood adversity, history of depression and anxiety, low self-esteem and even personality traits like resilience.[15 22–24] Despite this knowledge, detecting PND remains a challenge for the healthcare system, with one review finding that around 30%–70% of cases go undetected and only 15% receive adequate treatment.[25 26]

Current screening protocols include the Edinburgh Postnatal Depression Scale (EPDS)[27] during postpartum visits to assess risk of PND.[28] However, early detection of PND has remained challenging for many reasons, including inconsistencies in screening practices,[29] and failure to distinguish depressive symptoms due to their overlap with typical somatic experiences in the early postpartum period.[30] Furthermore, women may hesitate to seek care possibly because of the depression itself, but also stigma and fear of being judged as an imperfect mother, as well as concerns about antidepressant use during pregnancy and breast feeding.[31] Besides, in-clinic screening frequently relies on retrospective self-reports of diagnostically relevant information, making it susceptible to errors and biases associated with autobiographical recollection.[32]

Unquestionably, more efficient and effective methods for predicting PND in mothers are needed to enable early identification and intervention, thus improving prognosis, and reducing the burden of disease.[33] Previous studies that have attempted to develop predictive models of maternal depression primarily focus on the postpartum period only.[34–39] Few have used social media fingerprints[40–42] or biomarkers[36 43] in their models, and these studies largely depended on psychometric self-reports and limited modalities. These drawbacks compromise the predictive power of the models and illustrate why multivariate, real-time and unobtrusive approaches to data collection and symptom monitoring must be encouraged to develop better predictive models.

## Preterm birth

Among somatic pregnancy complications, PTB is a major cause of neonatal death, as well as of poor long-term health in children, affecting approximately 15 million babies worldwide each year.[44 45] In Sweden, the PTB rate is about 6%,[46] which is a relatively low number compared with the international average of over 10%.[44] Like PND, the aetiology of PTB is multifactorial, including previous PTB, multifetal pregnancy, cervical insufficiency, intrauterine infections, vaginal bleeding in the second trimester, in-vitro fertilisation, primiparity, as well as maternal antenatal stress and depression.[47–50] In fact, many risk factors overlap between PND and PTB, such as childhood traumatic events or maltreatment, stressful life events, being single or lacking social support, being overweight, smoking and low socioeconomic status.[51 52] Inflammation has been suggested as a possible underlying pathway for both depression and preterm delivery.[53]

There are evidence-based interventions for preventing or delaying PTB to optimise birth outcomes, such as smoking cessation, progesterone therapy, cerclage in women with cervical insufficiency or antibiotics.[51 54] However, a major obstacle for the success of these interventions is the aetiological heterogeneity of PTB, which makes it extremely challenging to identify women at high risk. In fact, two-thirds of women who experience PTB do not present with any risk factors at all.[55] Available biological diagnostic tests for PTB (such as fetal fibronectin) lack sufficient positive prediction values.[56] Screening for cervical length is performed in Sweden for women with a history of PTB; however, this is not helpful in primiparous women.[57]

It can be concluded that no single biomarker is sufficient for prediction; multimodal data, including psychosocial and behavioural factors, should, therefore, be the focus of prediction efforts.

## Digital phenotyping and big data

Digital devices like smartphones allow us to capture moment-by-moment, objective data regarding the patient's experiences and functions in non-clinical settings. This process, known as *digital phenotyping*,[58] allows us to collect two kinds of data: *active data* and *passive data*. Active data refer to data that require user input, such as surveys and voice recordings. Passive data refer to automatically collected data from smartphone sensors and activity logs, which can be used to infer the user's mobility and sleep patterns, digital social activity, smartphone usage patterns, and even affective and cognitive changes.

The Mom2B smartphone app is developed using the Beiwe research platform (www.beiwe.org) from the Harvard School of Public Health. It can allow us to capture digital phenotyping data during the perinatal period with greater efficiency and temporal sensitivity as data collection occurs continuously and in real-world contexts, which minimises the risk of recall biases. Such apps could also be integrated into the mother's perinatal care plan. One drawback of smartphone-based data

collection, in general, is that attrition rate increases with longer follow-up times[59]; however, this can at least in part be compensated for by the continuous collection of passive data.

In fact, one of the biggest advantages of smartphone-based digital phenotyping is the ability to collect multivariate, high-volume data, known commonly as big data.[60] Big data are excellent for healthcare research since it can facilitate a unique insight into risk factors and the development of better diagnostic frameworks[61]; however, the literature on big data approaches for psychiatric conditions, particularly perinatal mental health, is limited.[62] Nordic countries are in the forefront in this respect—with the Danish National Birth Cohort,[63] the Norwegian Mother, Father and Child Cohort (MoBa)[64] and Autism Birth Cohort (ABC)[65] studies, and the Swedish Biology, Affect, Stress, Imaging and Cognition (BASIC) cohort study[66]—due to the availability of nationwide registers with comprehensive personal and medical information for all pregnant women in these countries. Nonetheless, register data, while valuable, lack the multeity, continuity and veracity offered by digital phenotyping.[67] Furthermore, studies derived from these cohorts have largely relied on traditional statistical methods, which are limited in their ability to scale to large data sets and identify more subtle patterns in data.[68]

To date, few studies have applied digital phenotyping for prediction of psychiatric conditions, such as relapse in schizophrenia[69] and severity of mood episodes in bipolar disorder.[70] In the context of PND, while smartphone apps are widely used, their application has been largely focused on screening and intervention.[71] Only two studies have applied digital phenotyping for predicting PPD.[34 72] While these studies have reported encouraging results, their predictive ability is compromised due to limited modalities (using only active data in the form of questionnaires), infrequent measurement points and usage of more traditional statistical methods. The Mom2B study combines nationwide health and pregnancy register data with active and passive data collected through smartphone-based digital phenotyping to objectively monitor indicators of PND in non-clinical contexts.

In order to harness the full potential of big data, more advanced analytical methods, such as machine learning (ML) and deep learning (DL), are ideal. ML is an artificial intelligence approach that refers to various methods of enabling an algorithm to identify and learn intricate patterns in data to predict outcomes.[73] Modern ML methods, such as deep neural networks (DNNs), are uniquely suited to analysing big data sets as they can detect complex, high-dimensional interactions and structured information, without guidance, that can then be used to train predictive algorithms. DNN models are comprised of multiple 'hidden' processing layers, inspired by biological neural networks, consisting of a series of interconnected nodes that resemble neurons.[74 75]

Over the last decade, there has been a steady increase in the use of DL methods in medicine.[76] However, few studies have used ML for diagnosis or risk assessment in psychiatry, and those that do are often limited by modest sample sizes and modalities, or from using only traditional ML techniques.[73 77] To our knowledge, Mom2B is the first study to adopt a big data approach and use multimodal digital phenotyping with advanced ML techniques to develop predictive algorithms for PND and PTB.

## Objectives

Using large-scale, multimodal data collected through the Mom2B smartphone app, together with health and pregnancy information from national registers, the primary aim of this study is to assess the accuracy of advanced ML and DL methods in predicting development of PND (1) in the third pregnancy trimester, using data from the first trimester, and (2) during the early and late postpartum period, using data collected throughout pregnancy and childbirth.

A secondary aim of this study is to apply advanced ML and DL techniques using the multimodal data set to predict the risk of PTB.

## METHODS AND ANALYSIS
### Cohort description

Mom2B (www.mom2b.se) is a national ongoing smartphone app-based study; the app was launched at the end of November 2019 to App Store and Google Play. All Swedish-speaking women above the age of 18 owning a smartphone, who are either pregnant or within 3 months postpartum, are eligible to participate by registering and providing consent in the Mom2B app. Participating women are also asked for optional consent to be contacted for additional research studies within and from outside the Mom2B project (see online supplemental appendix A). Participant data are then linked to psychiatric and somatic health-related and pregnancy-related information available from Swedish national registers.

We aim to recruit at least 5000 participants with completed outcome measures. Due to the complexity of ML methods, it is not possible to perform any traditional test of statistical power. However, based on previous studies[59 78 79], and conferring with experts in artificial intelligence, we estimated that this approximate number would give us enough material to build robust prediction models while accounting for attrition and the prevalence rates of the outcomes.

Information about the study is being disseminated on social media, and through posters and brochures sent to primary and maternal care centres across the country. Figure 1 illustrates Mom2B recruitment, data collection and opt-outs. Table 1 outlines participant characteristics of the existing Mom2B cohort based on users who have contributed relevant data, along with similar characteristics in the general Swedish population.

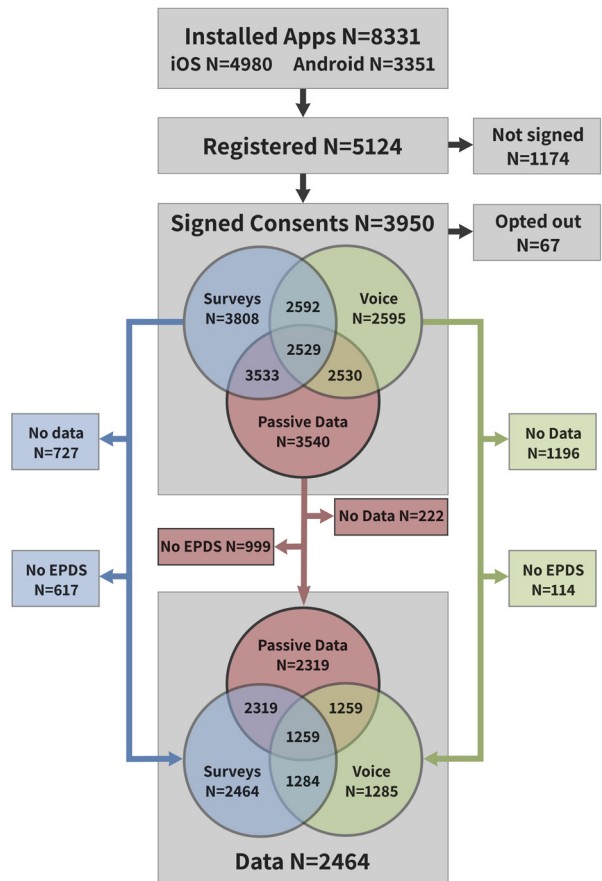

**Figure 1** From top to bottom, the grey content blocks in the main column represent installed apps (downloads of the Mom2B app by unique users from either App Store (iOS) or Google Play (Android)), registered users (individuals who have submitted registration information in the app), signed consents (registered users who have consented to contributing data, and signed these consents electronically) and, finally, data (participants with signed consents who, at minimum, have completed the Edinburgh Postnatal Depression Scale (EPDS[27]) at least once). The latter two blocks also illustrate the signed consents and available data, respectively, by type of data (survey, voice and passive data). The intersections of the Venn diagrams are non-exclusive, meaning that the number count in the intersection of surveys and passive data, for example, can include individuals who have also contributed to voice recordings. This flow chart reflects data last downloaded on 6 September 2021.

## Data collection

The Mom2B app collects three types of data: survey data, audio recordings, as well as passive data. Data can be collected from the first week of pregnancy, and up until week 52 after birth. Only data that participants have consented for are collected from the time they register to the study, and they can change their consent preferences anytime in the app if they wish to stop.

## Surveys and questionnaires

The Mom2B app delivers a range of both validated and self-developed questionnaires two to three times per week on average, with a mean of five questions per survey. These questionnaires are used to collect information regarding the participant's mental and physical well-being, and history, personality, relationships, as well as perinatal and parenthood experiences. They include the EPDS, a 10-item self-report screening tool with good psychometric properties,[80 81] used as the primary outcome measure in this study to assess depressive symptoms throughout the study period. A summary of the timeline of validated and self-developed instruments, along with the number of occurrences of the survey throughout the study period, can be found in figures 2 and 3, respectively.

## Voice recordings

Voice acoustic qualities, such as pitch, speed, timing and timbre, have been used in previous research to successfully distinguish depressed from non-depressed individuals.[82–85] To collect voice data, the Mom2B app sends out a voice recording task asking the participant to record reading simple texts, numerical sequences or vocalisations every 2–4 weeks.

## Passive data collection

Passive data that the user has provided consent for are continuously collected via the Mom2B app throughout the study period, and are used to infer the user's behavioural patterns. Some of these features are collected differently for iOS and Android users. The feature modalities are briefly explained below.

### Mobility

Correlations have been demonstrated between a patient's geographical movement patterns and changes in depressive symptoms.[86 87] Sixty seconds of Global Positioning System (GPS) data are continuously collected every 10 min. Accelerator data are collected when the motion exceeds a certain threshold; motion activities, including being stationary, walking, running, cycling and movement in a vehicle, are recorded when the state changes. For iOS, we collect device motion to provide more detailed motion sensor data.

### Data usage

Internet usage is the main feature of data usage. Various patterns of internet usage have been identified in relation to depression in different populations,[88 89] but not among women with PND. The Mom2B app records the accumulated upload and download rates together with timestamps. Another feature, reachability, records timestamped smartphone connectivity—whether the phone is connected to cellular network, Wi-Fi or neither. It also records a Wi-Fi log for Android phone, which includes anonymised Media Access Control (MAC) addresses' frequencies and Received Signal Strength Indicator (RSSI) of available wireless networks in the area.

### Smartphone usage

General smartphone use has been found to correlate with sleep quality and depression.[90 91] Phone power state, combined with mobility parameters, can reflect individual

**Table 1** Sociodemographic characteristics, pregnancy history and birth outcomes on participants in the Mom2B study and the general population of pregnant women in Sweden

| Characteristics | Mom2B (2020–2022) (n=3909)* | | | Sweden (2019)† | |
| --- | --- | --- | --- | --- | --- |
| | Available data (n) | Missing data (n) | n (%) or mean±SD | Available data (n) | n (%) or mean |
| Maternal age (years) | 3430 | 479 | 31.2±4.4 | 113 816 | 30.7 |
| Country of origin | 3441 | 468 | | 112 530 | |
| Sweden | | | 3177 (92.3) | | 78 033 (69.3) |
| Nordic countries except Sweden | | | 40 (1.2) | | 1280 (1.1) |
| Europe except Nordic countries | | | 116 (3.4) | | 9172 (8.2) |
| Outside Europe | | | 108 (3.1) | | 24 045 (21.4) |
| Education | 3444 | 465 | | 107 711 | |
| ≤12 years | | | 744 (21.6) | | 48 793 (45.3) |
| Post-secondary education | | | 2700 (78.4) | | 58 918 (54.7) |
| Employment before pregnancy | 1677 | 2232 | | 113 147 | |
| Working/student/ parental leave | | | 1626 (97) | | 103 967 (91.9) |
| Unemployed/sick leave | | | 51 (3) | | 9180 (8.1) |
| Smoking 3 months before pregnancy | 3041 | 868 | 441 (14.5) | 110 991 | 11 765 (10.6) |
| BMI before pregnancy (kg/m$^2$) | 3353 | 556 | 25.5±5.3 | 108 929 | |
| <18.5 | | | 70 (2.1) | | 2783 (2.5) |
| 18.5–25 | | | 1815 (54.1) | | 59 384 (54.6) |
| 25–<30 | | | 923 (27.5) | | 29 636 (27.2) |
| ≥30 | | | 545 (16.3) | | 17 126 (15.7) |
| Primiparous | 3268 | 641 | 1188 (36.4) | 113 816 | 48 473 (42.5) |
| Caesarean section | 1356 | 639‡ | 238 (17.5) | 114 757 | 20 312 (17.7) |
| Preterm delivery (<week 37) | 3311 | 598 | 190 (5.7) | 116 071 | 6502 (5.6) |

Percentages are given in relation to available data from women.
*Data downloaded on 1 February 2022.
†Data retrieved from the Swedish Medical Birth Register and Swedish National Board of Health and Welfare from 2019.
‡Calculated using the confirmed number of women in the postpartum period only.
BMI, body mass index.

behaviour like sleep patterns.[92 93] To keep track of the use of the smartphone device, data are collected on screen activity, charging status and device reboot.

### Social media activity

Social media behaviour has also been proven useful in detecting mental states. It has also been shown that reduced social activity on Facebook predicted symptoms of PPD.[41] Collected data consist of simple behavioural measures, such as posting, commenting or liking, together with their timestamps. Notably, we only measure activity levels, not information related to the text or image content of that activity, and participants are made aware of this when providing consent.

### Survey metadata

App-based surveys make it possible to also collect metadata. This kind of behavioural metadata may contain clinically relevant information related to attention, processing speed and working memory capacity, and even any deterioration of psychiatric symptoms.[58 94] We collect data on the time a survey was opened, time taken to answer each question and fully complete a survey, as well as any changes made in survey responses.

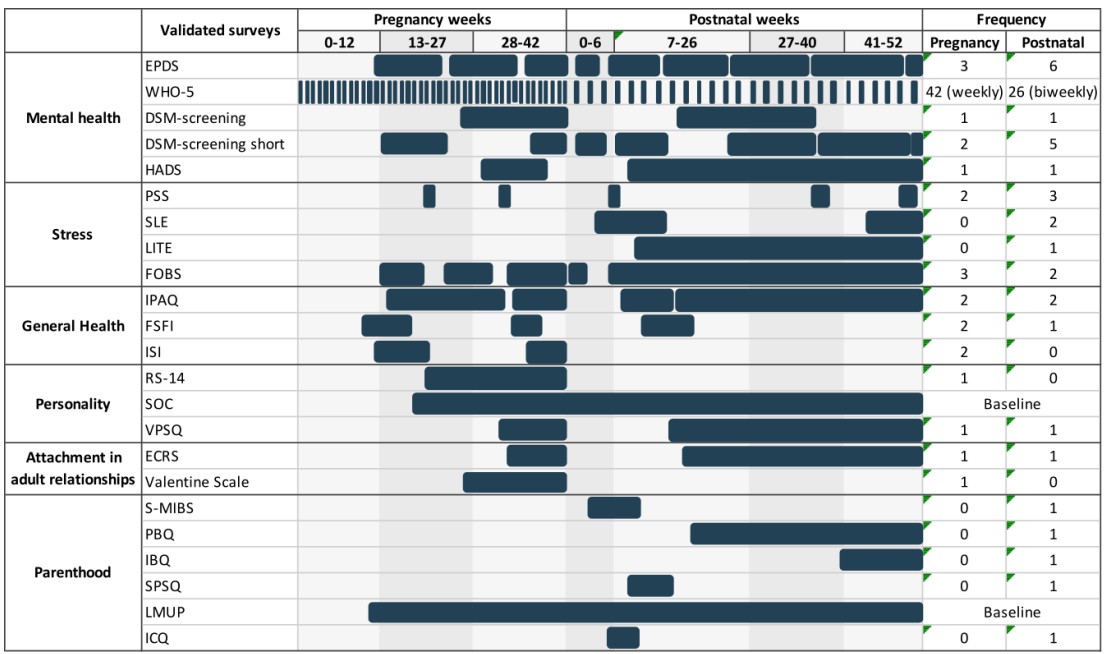

**Figure 2** Timeline of validated instruments administered in the Mom2B study during pregnancy and postpartum periods, and the number of occurrences for each instrument throughout the study period. Surveys become available to users for varying periods of time and will disappear once completed or when their period of availability is over. EPDS, Edinburgh Postnatal Depression Scale[27]; WHO-5, WHO-5 Well-Being Index[106]; DSM-screening, Diagnostic and Statistical Manual of Mental Disorders, 5th Edition, criterion for depression; DSM-screening short is a shortened version of the DSM-screening with selected questions chosen by the research team; HADS, Hospital Anxiety and Depression Scale[107]; PSS, Perceived Stress Scale[108]; SLE, Stressful Life Events[109]; LITE, Lifetime Influence of Traumatic Experiences[110]; FOBS, Fear of Birth Scale[111]; IPAQ, International Physical Activity Questionnaire[112]; FSFI, Female Sexual Function Index[113]; ISI, Insomnia Severity Index[114]; RS-14, Resilience Scale[115]; SOC, Sense of Coherence[116]; VPSQ, Vulnerability Personality Style Questionnaire[117]; ECRS, Experience in Close Relationships Scale[118]; Valentine Scale (relationship with your partner)[119]; S-MIBS, Swedish Mother to Infant Bonding Scale[120]; PBQ, Postpartum Bonding Questionnaire[121]; IBQ, Infant Behavior Questionnaire[122]; SPSQ, Swedish Parenthood Stress Questionnaire[123]; LMUP, London Measure of Unplanned Pregnancy[124]; ICQ, Infant Characteristics Questionnaire.[125]

**Figure 3** Timeline of self-developed surveys administered in the Mom2B study during pregnancy and postpartum periods, and the number of occurrences for each instrument throughout the study period. Surveys become available to users for varying periods of time and will disappear once completed or when their period of availability is over.

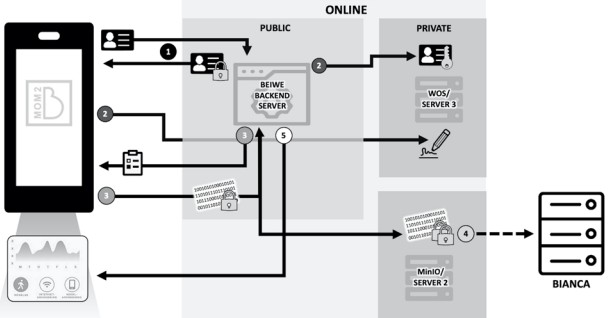

**Figure 4** Flow of data from user to servers for storage and analysis. Data pass through secure servers accessible only by authorised members of the Mom2B team, and can be decrypted for analysis in Bianca when needed.

### National register data

Supplementary information will be accessed via the following Swedish national health and quality registers: the Medical Birth Register, the Pregnancy Register, the National Patient Register, the Prescribed Drug Register and population censuses from Statistics Sweden. The accessed information includes records of perinatal complications such as PND, PTB or any other complications considered important risk factors for the study outcomes, such as gestational diabetes, gestational hypertension, pre-eclampsia, prolonged delivery, severe lacerations, postpartum haemorrhage, induction of delivery, instrumental vaginal delivery, caesarean section and small for gestational age.

Further, the mother's weight at enrolment in maternity care and at aftercare visits; calculated date of birth from last menstrual period and from ultrasound; and information on previous miscarriages, previous abortions, chronic diseases, fear of childbirth, involuntary infertility, gestational age at enrolment in maternity care and fetal diagnostics will be obtained from the Medical Birth Register. Retrieved information also includes variables regarding the background, health and lifestyle of the participant for validation purposes of our self-report questionnaires, as well as psychiatric and somatic morbidity for up to 15 years after childbirth.

### Data flow and storage

Figure 4 illustrates the data flow and storage process as follows:

► Participants register to the study via the Mom2B app using their Swedish Social Security number, which is encrypted in the device using a private key provided by the Beiwe backend server, and replaced by a random, pseudoanonymised code number.

► The decryption key, together with the participant consent information and electronic signatures, is stored in a private, write-only server at Uppsala University.

► The app fetches surveys and voice recording tasks, and uploads data from participants to the backend server, where it is encrypted and sent to MinIO, a

secure, cloud-based storage, where another layer of encryption is added. Passive data collected from the phone follow the same path.

► From MinIO, all data are sent to Bianca, a private offline server, in both encrypted and decrypted forms for storage and analysis, respectively.

► The app provides weekly reports based on participant activity and fetches health-related information relating to the perinatal period the user is in, as well as frequently asked questions about the study and perinatal health.

### Preliminary data analysis strategy

The Mom2B data set contains different modalities, including audio data, sensor data and survey data, which will be analysed separately and then combined. We plan to use both traditional ML and DL techniques in order to determine reliable predictors of PND and develop accurate predictive algorithms, and will report our findings following the best fit current guidelines, such as the Transparent Reporting of a Multivariable Prediction Model for Individual Prognosis or Diagnosis (TRIPOD) statement.[95]

### Feature engineering

To handle these multimodal data, we will extract features for these modalities separately using traditional feature engineering as well as DL techniques.[96 97] An example of traditional feature engineering uses trajectories,[87] which use mobility features, such as number of significant places visited, maximum distance and SD. However, DL can be used for extracting features in many other modalities, such as social media and audio data, which are not investigated widely in the area.

### Feature selection and model selection

To analyse the multimodal Mom2B data set, we will start with each modality separately. To reduce the possibility of potential overfitting, given the numerous features in our data set, we will use recursive feature elimination to obtain the optimal set of variables for further model development.

Previously, logistic regression, support vector machine, random forests, XGBoost and neural networks have been the most commonly used and efficient ML algorithms for prediction of PND.[98] An advantage of using such traditional ML methods is to give us a feature importance ranking, allowing us to identify stronger predictors. Using DL to analyse digital phenotyping data for evaluating risk of depression is a relatively novel approach compared with traditional ML models.[73] DL models have been shown to outperform traditional ML in various tasks involving complex data sets,[99 100] and can be combined with traditional ML in multimodal data mining tasks to further improve performance.[97] We will test and select the best performing ML models for each modality and determine strong predictors of PND.

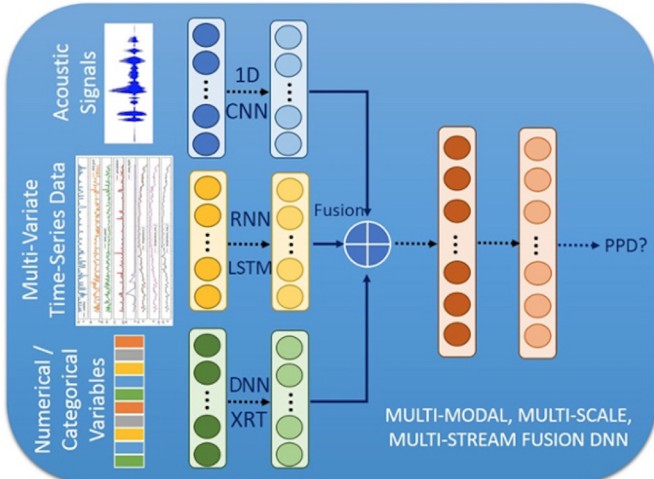

**Figure 5** A multimodal machine learning model for peripartum depression (PPD) diagnosis. The extracted features can be classified into three categories: acoustic signals, time series features and categorical features. We can then determine the most suitable model for each category. For example, for acoustic signals, we would apply convolutional neural network (CNN); for time series data, we would apply recurrent neural network (RNN) such as long short-term memory (LSTM); and for numerical variables, we would apply deep neural networks (DNNs) such as transformers, or traditional models like extremely randomised trees (XRT), gradient boosted trees, etc. These models can yield high-dimension representations of multimodal features. After feature fusion, the integrated features will be fed into another neural network for prediction.

### Multimodal computational model

The multimodal data we collect are in different scales, dimensions and formats, which need to be harmonised before prediction.[101] Different models are better suited to perform on different modalities. To handle this complexity in multiple data modalities, we consider modality fusion during the development phase.

One example of a multimodal ML model is shown in figure 5. The model is designed to detect potential depressive episodes based on multimodal data collected in the Mom2B app. Preprocessed data in three modalities are fed into models and intermediate representations are then fused together and fed in as input features of a classification model.

### Evaluation metrics

Our data will be split into a training data set, for analysis, and a test data set, to assess model performance. We consider using multiple evaluation metrics including area under the receiver operating characteristic curve, specificity, sensitivity, positive predictive value, negative predictive value, balanced accuracy and F1 score, as these measurements vary in importance according to the setting and goal of the final algorithm. Thereby, we can compare the performance of traditional ML with DL, and the different assemblies of models, from different perspectives in the context of prediction of PND.

### Patient and public involvement

A qualitative study is planned for exploring the attitudes and concerns of participating women towards the Mom2B app. Furthermore, an online survey will be sent to women who have had no recent activity on the app or withdrawn participation from the study. Direct contact with end users and the ability to make changes to the app based on their feedback can enhance user experience and increase engagement. A representative from Mamma till Mamma, a non-profit organisation in Sweden focused on perinatal mental well-being, serves on our advisory board. The organisation has been involved in the piloting of our study and design of questionnaires, and currently supports us with recruitment. We plan to involve them in the dissemination of study results as well.

### Substudies

In addition to predicting PND and PTB, the rich data collected from the Mom2B cohort will also be used to investigate further questions, mainly regarding the health of pregnant and postpartum women. Other planned areas of research are regarding the impact of early mother–infant separation and neonatal intensive care of the baby on the well-being of the mother; and sexual function and its potential correlates to depression and anxiety in the perinatal period.

### Maternal depression and well-being during the COVID-19 pandemic

In the beginning of 2021, data collected from 1577 participants were used to assess depressive and anxiety symptoms, as well as well-being and life changes in pregnant women in Sweden during the COVID-19 pandemic (from February 2020 to March 2021).[102] The Mom2B app enabled gathering psychiatric information at a national level during the pandemic, as well as passive data on mobility. Levels of perinatal affective symptoms and low well-being were elevated compared with previous years and to months with fewer cases. Similar apps can help healthcare providers and governmental bodies to monitor high-risk groups during crises in real time, as well as to adjust measures and the support offered.

## ETHICS AND DISSEMINATION

Participants are informed about the aims of the study, and that the confidentiality and security of their data will be assured. All participants provide their consent to participate while registering to the study, and are informed that they can withdraw their participation at any time without giving a reason. Ethical approval has been obtained from the Swedish Ethical Review Authority (dnr: 2019/01170, with amendments) and the project fulfils General Data Protection Regulation (GDPR) requirements, including the processing, storage and protection of all data. Results will be continuously disseminated through international peer-reviewed journals, the project's website and social media channels, and presented in national and

international conferences. All publications will be open access.

## DISCUSSION
### Strengths and limitations

Besides the utility of digital phenotyping in combination with the advanced analytical methods planned to be used, other strengths of the Mom2B study include the involvement of participants. Statistics based on the WHO-5 Well-Being Index and behavioural data (movement, internet usage, sleep, etc) collected from participants are sent to the user, allowing them to follow their well-being and activity as an incentive for continued participation. Weekly informational reports regarding common experiences and concerns for both the mother and the child for that particular week of the perinatal period, based on information taken from 1177.se (Swedish healthcare service), are available to users and allow them to easily stay informed. As per standard guidelines,[103] if participants receive a high score on the EPDS, they are prompted to contact their healthcare provider or emergency support services for support, and if unsure, they can contact the research team, which will help them find appropriate support for their needs. Continuous contact is maintained with participants until they find support.

The involvement of user organisations and an international advisory board further strengthens the study by increasing the feasibility, the use of state-of-the-art methods and the potential for high acceptance by the end users, which is especially important for future integration in regular clinical practice.

However, there are some limitations to acknowledge. Weekly reports and statistics are important in supporting and incentivising users, but it is possible they may influence users' responses to certain questionnaires. To account for this, we consider including how often they are checked by users as a feature within our models. Furthermore, our app is available only in Swedish, which excludes a number of otherwise eligible participants, and the high costs for maintaining the technical infrastructure in the frontend and backend of the app require considerable funding. Attrition is also an issue, especially with data that require active input from users. While we can attempt to combat this by improving the app based on user feedback, it is important to consider that attrition might also reflect the worsening of symptoms and be a predictor per se of clinical deterioration. It will be important to distinguish such participants and determine how to use attrition as a predictor variable.

### Future perspectives

We are at the beginning of the smartphone-based research era, and future possibilities seem numerous. We intend to develop the Mom2B app in other languages, including English, to expand to a more diverse and wider population. If the app succeeds in developing good predictive models for PND, the research team anticipates that the app could be further developed to include evidence-based interventions.[104] Furthermore, since PND is much less understood in co-parents and improving the other parent's mental well-being is conducive to the health of the mother and the children as well,[105] the app could be further developed to study co-parental PND. The Mom2B research team plans to further adapt the app to other research topics such as teenage and student mental health, and prediction of new episodes or self-harm in major depression.

**Author affiliations**
[1]Department of Medical Sciences, Uppsala University, Uppsala, Sweden
[2]Centre for Women's Mental Health during the Reproductive Lifespan (Womher), Uppsala University, Uppsala, Sweden
[3]Department of Women's and Children's Health, Uppsala University, Uppsala, Sweden
[4]Centre for Translational Microbiome Research, Department of Microbiology, Tumor and Cell Biology, Karolinska Institute, Stockholm, Sweden
[5]Department of Information Technology, Uppsala University, Uppsala, Sweden

**Acknowledgements** We would like to extend our gratitude to Oskar Burman for creating and continuously adjusting the smartphone research application Mom2B; to Dr Masoumeh Rezapour, the previous head of Department of Obstetrics and Gynecology at Uppsala University Hospital, for support and guidance in the first steps of this project; to Arvid Marklund and Stavros Iliadis for support at the early stages of the app creation; and to Maria Grandahl, Emilia Biskop and Lisa Lindgren for their valuable input regarding app content and questionnaires.

**Contributors** AS and FCP were involved in the conception and design of the study, revised the draft critically for intellectual content and approved the final manuscript. EB, EF and CA contributed significantly to the design and launch of the project, and revised the draft critically for intellectual content. AE, EB, UE, MZ and KG contributed to the drafting of the protocol and the continuation of the project. AMB is the coordinator of the study and was primarily responsible for drafting the manuscript and revising it critically for intellectual content.

**Funding** This work was supported by the Uppsala Region to AS, the Swedish Association of Local Authorities and Regions (SKR) to the Department of Obstetrics and Gynecology, Uppsala University Hospital, the Swedish Research Council (grant number 2020-01965) to AS, as well as the Swedish Brain Foundation and Uppsala University Womher School.

**Competing interests** None declared.

**Patient and public involvement** Patients and/or the public were involved in the design, or conduct, or reporting, or dissemination plans of this research. Refer to the Methods section for further details.

**Patient consent for publication** Not required.

**Provenance and peer review** Not commissioned; externally peer reviewed.

**ORCID iD**
Ayesha M Bilal http://orcid.org/0000-0002-7349-8765

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
