## [Reviewer comments · BMJ Open]

ARTICLE DETAILS

TITLE (PROVISIONAL)	Predicting perinatal health outcomes using smartphone-based digital phenotyping and machine learning in a prospective Swedish cohort (Mom2B): study protocol
AUTHORS	Bilal, Ayesha; Fransson, Emma; Bränn, Emma; Eriksson, Allison; Zhong, Mengyu; Gidén, Karin; Elofsson, Ulf; Axfors, Cathrine; Skalkidou, Alkistis; Papadopoulos, Fotios C.

VERSION 1 – REVIEW

REVIEWER	Kaliush, Parisa University of Utah, Clinical Psychology
REVIEW RETURNED	20-Jan-2022

GENERAL COMMENTS	I attached a Word document of this review to my submission. Thank you for the opportunity to review this article. Overview This protocol paper describes an innovative and impactful project, the Mom2B Study, that aims to use active and passive data collection in order to predict perinatal depression and preterm birth among a large-scale national sample of Swedish-speaking women. Given their use of digital phenotyping and “big data” methods, the authors plan to employ machine learning and deep learning analytic techniques in order to develop predictive algorithms for perinatal depression and preterm birth. The questions, comments, and suggested revisions below are intended to strengthen this protocol paper for possible publication. Data Collection 1. Tables 2 and 3 offer helpful visuals for understanding which surveys will be administered and when. However, it is unclear how survey administration decisions were made. For example, why is the EPDS administered multiple times during postpartum weeks 0 – 27 but only once during weeks 28 – 40 and once during weeks 40 – 52? Recent research indicates that postpartum women in the United States may be more likely to die by suicide and/or drug overdose between postpartum months 9 – 12 (e.g., Mangla et al., 2019), so it may be worth considering more frequent EPDS data collection during the later postpartum time points.2. It would be helpful to know more details about the surveys listed in Table 3. For instance, what surveys will be administered to measure sleep and breastfeeding? Are these measures validated or self-developed? Without this information, it seems as if this study cannot easily be repeated.3. Given that the authors plan to collect voice recording data, have they considered collecting recordings of infant cry vocalizations? There has been some interesting machine learning-based
--

	research on infant cry vocalizations and potential associations with postpartum depression (e.g., Gabrieli et al., 2020; Orlandi et al., 2016; Osmani et al., 2017). 4. Are passive data being collected continuously from pregnancy through 52 weeks postpartum? Clarification on this design consideration would be helpful. 5. Given that the authors are planning to collect data on women's experiences with previous miscarriages, abortions, and feelings about childbirth, they may consider assessing if the present pregnancy was unplanned and/or unwanted, as unplanned pregnancy is a risk factor for postpartum depression (e.g., Faisal-Cury et al., 2017; Yanikkerem et al., 2012). Strengths and Limitations 1. It would be helpful if the authors provided more details about "the involvement of participants" in their study (p. 11). For instance, what mental health information is included in the reports that are sent back to participants? How often do participants receive these reports? Are the participants made aware that these reports are based on research and not professional clinical assessment? If participants contact the research team for mental health support, what protocol do the authors follow? What resources are the authors using to inform these ethical decisions? 2. Relatedly, I wonder if it is possible that regular reports about participants' mental health and digital phenotyping data may influence ongoing data that they provide. Have the authors considered this possibility? If so, how do they plan to account for these effects? Minor Comments 1. The authors are encouraged to review their manuscript for occasional grammatical errors (e.g., "the app could be further developed to include evidence-based interventions interventions" [p. 12]). 2. Without the in-text information under Data Flow and Storage, Figure 2 is challenging to understand. If Figure 2 will not be placed next to this in-text section upon publication, the authors are encouraged to move this information to the Figure 2 caption. References Faisal-Cury, A., Menezes, P. R., Quayle, J., & Matijasevich, A. (2017). Unplanned pregnancy and risk of maternal depression: Secondary data analysis from a prospective pregnancy cohort. Psychology, Health and Medicine, 22, 65-74. Gabrieli, G., Bornstein, M. H., Manian, N., & Esposito, G. (2020). Assessing mothers' postpartum depression from their infants' cry vocalizations. Behavioral Sciences, 10, 1-10. Mangla, K., Hoffman, C., Trumpff, C., O'Grady, S., & Monk, C. (2019). Maternal self-harm deaths: An unrecognized and preventable outcome. American Journal of Obstetrics and Gynecology, 221, 295-303. Orlandi, S., Reyes Garcia, C. A., Bandini, A., Donzelli, G., & Manfredi, C. (2016). Application of pattern recognition techniques to the classification of full-term and preterm infant cry. Journal of Voice, 30, 656-663. Osmani, A., Hamidi, M., & Chibani, A. (2017). Machine learning approach for infant cry interpretation. International Conference on Tools with Artificial Intelligence, 182-186. Yanikkerem, E., Ay, S., & Piro, N. (2012). Planned and unplanned pregnancy: Effects on health practice and depression during pregnancy. Journal of Obstetrics and Gynecology Research, 39, 180-187.
--	---

REVIEWER	Heaukulani, Creighton Ministry of Health, Office for Healthcare Transformation
REVIEW RETURNED	28-Jan-2022

GENERAL COMMENTS	I'm glad to see this study. The application of digital phenotyping to this population is likely to be highly impactful, and its investigation here is very timely. What's more, the size of the cohort and in particular the development of the Mom2B App covering both iOS and Android, and its distribution to so many phones, are impressive. The authors should be proud of these accomplishments. I request that the data analysis section be expanded to include specific plans for linear modelling and extraction of corresponding inferences. I understand that the authors are leaving much of the deep learning modelling to exploration, and this is great! But you should also have plans for linear models (hierarchical/multi-level/random effects models). In my opinion, these should always be carried out and studied before playing around with deep learning models anyway, which are usually only useful for prediction. Showing inferences from the linear models will standardize the results across the literature and will most likely produce very interesting insights. I do not think conducting this analysis would be hard or overly time consuming, and I believe it will help produce insights to guide how you eventually construct your deep learning architectures or approach feature engineering. Perhaps the authors intended to do this when they said that "traditional ML" methods will be explored, but it should be made explicit. In the spirit of pre-declaration, the protocol should include very specific data analysis plans including a specific model and the way significance of inferences will be determined, how multiple comparisons are avoided, etc. Currently this does not exist in the protocol.
--

VERSION 1 – AUTHOR RESPONSE

Comments from Reviewer 1

Data collection

1. Tables 2 and 3 offer helpful visuals for understanding which surveys will be administered and when. However, it is unclear how survey administration decisions were made. For example, why is the EPDS administered multiple times during postpartum weeks 0 – 27 but only once during weeks 28 – 40 and once during weeks 40 – 52? Recent research indicates that postpartum women in the United States may be more likely to die by suicide and/or drug overdose between postpartum months 9 – 12 (e.g., Mangla et al., 2019), so it may be worth considering more frequent EPDS data collection during the later postpartum time points.

⇒ Thank you for this suggestion. Based on your comments, it is apparent that our tables lacked the clarity we were hoping for. We have replaced the Tables 2 and 3 with newly added Figures 2 and 3 that illustrate a more precise timeline of when the surveys are delivered, for how long they remain available for completion, and the number of times in total that they occur throughout the study period. We hope this addresses the matter satisfactorily.

⇒ Regarding the importance of assessing depression in late pregnancy, you raise an important point. However, the EPDS was developed as a tool to screen for depressive symptoms during early postpartum period, ideally in weeks 6-12 after birth. This is also the period when onset of symptoms is most common, which of course could persist and lead to severe outcomes such as

suicide later in the postpartum period. It has also been shown to be a valid measure of depression during pregnancy (Levis et al., 2020). We absolutely do agree with the need to continue screening in the late postnatal period, and while we deliver EPDS only twice, we also continue to monitor emotional wellbeing through the DSM5-short questionnaires and the WHO5 Wellbeing Index, the latter being delivered biweekly. Moreover, information will also be acquired from patient registers regarding diagnoses, psychiatric hospitalizations, and prescription of psychiatric medications, which will supplement our outcome measures even in the late postpartum period.

1. It would be helpful to know more details about the surveys listed in Table 3. For instance, what surveys will be administered to measure sleep and breastfeeding? Are these measures validated or self-developed? Without this information, it seems as if this study cannot easily be repeated.

⇒ Thank you for pointing this out. We agree that the tables may not have been sufficiently clear in describing the surveys used and the timeline. Moreover, we noticed that certain validated surveys had incorrectly been mentioned as self-developed questionnaires, and were missing from Table 2 and 3, which was an oversight on our part. As mentioned above, we have now replaced Tables 2 and 3 with Figures 2 and 3, respectively, that distinguish between validated and self-developed surveys. We have taken care to comprehensively list all self-developed surveys we use in terms of the construct/subject they assess. We hope the changes resolve the concerns for replicability and clarify the factors taken into account in our analysis.

1. Given that the authors plan to collect voice recording data, have they considered collecting recordings of infant cry vocalizations? There has been some interesting machine learning-based research on infant cry vocalizations and potential associations with postpartum depression (e.g., Gabrieli et al., 2020; Orlandi et al., 2016; Osmani et al., 2017).

⇒ Thank you for the suggestion. It would certainly be interesting to explore this as a future research topic, perhaps with a subset of women from the Mom2B cohort. However, in the case of the current study, there are some practical reasons why this may be too ambitious to accomplish in the ongoing cohort. Collecting data from infants has practical challenges (Ji et al., 2021); would require additional consent from the co-parent; and, by placing a greater demand on new mothers, we also risk a greater opt-out rate. However, given the ongoing status of the Mom2B cohort, we thank the Reviewer for bringing it to our attention, and we remain open to such a study after we assess its feasibility.

1. Are passive data being collected continuously from pregnancy through 52 weeks postpartum? Clarification on this design consideration would be helpful.

⇒ We have accordingly revised the sub-section 'Passive data collection' to clarify the nature of passive data collection with the amended statement: "Passive data that the user has provided consent for are continuously collected via the Mom2B app throughout the study period, and are used to infer the user's behavioral patterns". We have also added a brief sentence under the sub-heading "Data Collection" to clarify the periods and conditions under which data is collected, as follows: "Data can be collected from the first week of pregnancy, and up till week 52 after birth. Only data that participants have consented for is collected from the time they register to study, and they can change their consent preferences anytime in the app if they wish to stop". We hope this resolves the lack of clarity.

1. Given that the authors are planning to collect data on women's experiences with previous miscarriages, abortions, and feelings about childbirth, they may consider assessing if the present pregnancy was unplanned and/or unwanted, as unplanned pregnancy is a risk factor for postpartum depression (e.g., Faisal-Cury et al., 2017; Yanikkerem et al., 2012).

⇒ We agree that this is an important factor to consider. While we had been assessing pregnancy planning previously using a single, direct question, which had been missing in the previous tables listing the surveys, we have since incorporated the London Measure of Unplanned Pregnancy (Barrett et al., 2004), a validated survey to assess how planned the pregnancy was. The survey is now included in Figure 2 illustrating validated surveys.

Strengths and Limitations

1. It would be helpful if the authors provided more details about “the involvement of participants” in their study (p. 11). For instance, what mental health information is included in the reports that are sent back to participants? How often do participants receive these reports? Are the participants made aware that these reports are based on research and not professional clinical assessment? If participants contact the research team for mental health support, what protocol do the authors follow? What resources are the authors using to inform these ethical decisions?

⇒ We agree that information regarding the reports sent back to patients should have been elaborated. We have revised the ‘Strengths and limitations’ section to clarify exactly what information is sent back to the users, that is, statistics and weekly informational reports. The text now reads: “Statistics based on WHO-5 and behavioral data (movement, internet usage, sleep etc.) collected from participants are sent to the user, allowing them to follow their wellbeing and activity as an incentive for continued participation. Weekly informational reports regarding common experiences and concerns for both the mother and child for that particular week of the perinatal period, based on information taken from 1177.se (Swedish healthcare service), are available to users and allow them to easily stay informed”. We hope it is clear within the text now that the reports are sent on a weekly basis according to the perinatal week women are in.

⇒ We have added the phrase “based on information taken from 1177.se (Swedish healthcare service)” to clarify where the information in the informational reports is taken from, and this information is clearly stated at the end of each report sent to participants.

⇒ As for the lack of explanation of the protocol for women who score highly on the EPDS, we thank you for pointing that out. We have, accordingly, added this information, as well as the explanation that our protocol is per standard guidelines, citation included. The text now reads: “As per standard guidelines[103], If participants receive a high score on the EPDS, they are prompted to contact their healthcare provider or emergency support services for support, and if unsure, they can contact the research team, which will help them find appropriate support for their needs. Continuous contact is maintained with participants until they find support”.

1. Relatedly, I wonder if it is possible that regular reports about participants’ mental health and digital phenotyping data may influence ongoing data that they provide. Have the authors considered this possibility? If so, how do they plan to account for these effects?

⇒ We thank the Reviewer for raising this important point. There is, of course, a possibility that being frequently informed about their own mood state, as well as the pregnancy or postnatal period, and seeing a statistical summary of their behavioral data may act as an intervention of sorts and influence their responses on certain surveys. We believe nevertheless that this is an important ethical obligation from our side, as well as an incentive for continued participation, to continue to provide support and easy access to information regarding the perinatal period in general as well as their personal mood and activity. However, the reviewer’s comment has made us consider solutions for accounting for these effects. We are looking into acquiring app metadata regarding how often the user checked the weekly reports and statistics, and incorporating that information as a variable within our models. It is important to note that this model is generalizable in digital contexts, considering that even other variables, such as passive data, are only possible to acquire using some kind of digital device. This is also now included among the limitations of our study.

Minor Comments

1. The authors are encouraged to review their manuscript for occasional grammatical errors (e.g., “the app could be further developed to include evidence-based interventions interventions” [p. 12]).

⇒ Thank you for bringing this to our attention. We have carefully reviewed the draft again to identify any grammatical or spelling errors.

1. Without the in-text information under Data Flow and Storage, Figure 2 is challenging to understand. If Figure 2 will not be placed next to this in-text section upon publication, the authors are encouraged to move this information to the Figure 2 caption.

⇒ We completely agree. However, all figures, including Figure 2 (now labelled Figure 4) must be submitted as separate files to BMJ open, and are therefore not placed in the document submitted to BMJ. We assume they will be placed in proximity to the in-text section titled Data Flow and Storage upon publication.

Comments from Reviewer 2

I request that the data analysis section be expanded to include specific plans for linear modelling and extraction of corresponding inferences. I understand that the authors are leaving much of the deep learning modelling to exploration, and this is great! But you should also have plans for linear models (hierarchical/multi-level/random effects models). In my opinion, these should always be carried out and studied before playing around with deep learning models anyway, which are usually only useful for prediction. Showing inferences from the linear models will standardize the results across the literature and will most likely produce very interesting insights. I do not think conducting this analysis would be hard or overly time consuming, and I believe it will help produce insights to guide how you eventually construct your deep learning architectures or approach feature engineering. Perhaps the authors intended to do this when they said that “traditional ML” methods will be explored, but it should be made explicit. In the spirit of pre-declaration, the protocol should include very specific data analysis plans including a specific model and the way significance of inferences will be determined, how multiple comparisons are avoided, etc. Currently this does not exist in the protocol.

⇒ We thank the Reviewer for these thoughtful suggestions. We would like to just clarify that the primary aim of the current project is to develop prediction models (not inferential models). For this, we do plan to use, for e.g., logistic regression (among other machine learning methods), but not for the purpose of hypothesis testing and the corresponding extraction of inferences. The multilevel models the reviewer is suggesting are not part of our plan for the time being, as we instead plan to use traditional feature engineering and selection techniques, as well as DL techniques to construct our feature set.

⇒ We agree with the recommendation to elaborate our analysis plan. We revised the sub-section titled ‘Preliminary data analysis strategy’ to incorporate more explicit details regarding our analysis plan, including further details about how we will approach feature extraction, what models we plan to use, and evaluation criteria for our predictive models. We agree that it is important to highlight the variables of importance for implementation in healthcare settings (explainable AI) because healthcare professionals may not be prone to trusting and using the algorithms if they don’t understand them (black box problem), however, the importance of the variables will be based on their predictive values, not causal effects.

⇒ Beyond our primary aim with this data, it is of course true, as the Reviewer suggested, that the Mom2B project is going to generate a big and detailed dataset. Part of the purpose with this Study Protocol is to make our dataset known so that collaborations can be established and further sub-studies planned. Although we would optimally pre-specify all such plans, at this point we don’t have an overview of the sub-studies that the project may inspire. However, we fully agree with the importance of pre-specification of analysis plans (especially for confirmatory hypothesis testing). Given that the sub-studies will involve secondary use of already collected data, the preregistration would need to make it clear what access the authors have had to the data prior to writing the protocol. As such, full study protocols and analysis plans of these and other future sub-studies will optimally be registered prospectively to analysis, together with disclosure of prior knowledge about the study data. For exploratory and confirmatory hypothesis tests alike, we will make sure to report all the analyses made, to avoid a situation with selective reporting among multiple analyses. In sum, even if a Study Protocol such as this does not fulfill the same purpose as a preregistered protocol, we hope that it contributes to transparency by declaring what data will be available for secondary use.

Changes made in document

1. New sub-section titled 'Objectives' added in the Introduction section.
2. Restructuring of document sections.
3. Table 1 has been updated to reflect more recent Mom2B cohort characteristics.
4. Figure 1 has been updated to reflect more recent Mom2B cohort statistics.
5. Table 2 and 3 have been removed and replaced with Figure 2 and 3 respectively.
6. Figure 2 has been displaced and renamed as Figure 4.
7. Figure 3 has been displaced and renamed as Figure 5.
8. Appendix A has been added as a supplementary document detailing the patient consent form.
9. Changes and additions in text are indicated via the tracking feature on MS Word. Text crossed out and in red indicates text that was erased or moved to another location. Text in blue should indicate new text added. Text in green should indicate text that was moved from another location. Text crossed out and in green indicates text that was moved to another location. It should be noted, however, that occasionally these color codes may not always be correct as different manners of moving or editing text (retyping, cutting and pasting, copying and pasting, etc.) may have led to it being color coded differently.

References:

Levis, B., Negeri, Z., Sun, Y., Benedetti, A., & Thombs, B. D. (2020). Accuracy of the Edinburgh Postnatal Depression Scale (EPDS) for screening to detect major depression among pregnant and postpartum women: systematic review and meta-analysis of individual participant data. *bmj*, 371.

Ji, C., Mudiyansele, T. B., Gao, Y., & Pan, Y. (2021). A review of infant cry analysis and classification. *EURASIP Journal on Audio, Speech, and Music Processing*, 2021(1), 1-17.

Barrett, G., Smith, S. C., & Wellings, K. (2004). Conceptualisation, development, and evaluation of a measure of unplanned pregnancy. *Journal of Epidemiology & Community Health*, 58(5), 426-433.

VERSION 2 – REVIEW

REVIEWER	Kaliush, Parisa University of Utah, Clinical Psychology
REVIEW RETURNED	08-Apr-2022

GENERAL COMMENTS	The authors thoughtfully addressed my comments, and I am eager to learn more about the study via future publications. Thank you for the opportunity to assist with this review process.
---

REVIEWER	Heaukulani, Creighton Ministry of Health, Office for Healthcare Transformation
REVIEW RETURNED	11-Apr-2022

GENERAL COMMENTS	I have read the author responses and the updated manuscript. My comments should not hold up the publication of this protocol. I recommend acceptance. I would, however, like to seriously urge the authors to look into inferential studies, as you mention, from the start. I think that will do great service to the community and insights from those results will live long beyond any deep learning model you produce (in my opinion). I understand you expect future studies/analyses to be undertaken by yourself or others, but it seems like a bit of a waste for you not to produce this impact on your own when you've worked so hard to get all of this data. I do agree this dataset will make a terrific contribution.
--